# Disease Modifying Monoclonal Antibodies and Symptomatic Pharmacological Treatment for Alzheimer’s Disease

**DOI:** 10.3390/biomedicines12112636

**Published:** 2024-11-19

**Authors:** Xiaoming Qi, Damir Nizamutdinov, Song Stephen Yi, Erxi Wu, Jason H. Huang

**Affiliations:** 1Department of Neurology, Baylor Scott & White Health, Temple, TX 78508, USA; xiaoming.qi@bswhealth.org; 2Department of Neurosurgery, Neuroscience Institute, Baylor Scott & White Health, Temple, TX 78508, USA; damir.nizamutdinov@bswhealth.org (D.N.); song.yi@bswhealth.org (S.S.Y.); erxi.wu@bswhealth.org (E.W.); 3Department of Neurosurgery, Baylor College of Medicine, Temple, TX 76508, USA; 4College of Medicine, Texas A&M University, College Station, TX 77843, USA; 5College of Irma Lerma Rangel College of Pharmacy, Texas A&M University, College Station, TX 77843, USA; 6LIVESTRONG Cancer Institutes, Department of Oncology, Dell Medical School, The University of Texas at Austin, Austin, TX 78712, USA

**Keywords:** Alzheimer’s disease, amyloid beta, monoclonal antibodies, aducanumab, lecanemab, donanemab

## Abstract

Alzheimer’s Disease (AD) is an irreversible, progressive syndrome characterized by neurocognitive impairment. Two neuropathological features seen in AD are extracellular amyloid plaques consisting of amyloid beta1-40 and 1-42, and intracellular neurofibrillary tangles (NFTs). For decades, neuroscience research has heavily focused on seeking to understand the primary mechanism of AD and searching for pharmacological approaches for the treatment of dementia. Three monoclonal antibodies that act against amyloid beta—aducanumab, lecanemab, and donanemab—have been approved by the Food and Drug Administration (FDA) for the treatment of mild cognitive impairment and mild AD, in addition to medications for cognitive symptom management such as acetylcholinesterase inhibitors and the N-methyl-D-aspartate (NMDA) antagonist. Further trials should focus on the combination of therapies targeting amyloid plaques and tau pathology.

## 1. Introduction

Alzheimer’s Disease (AD) was first introduced by Alois Alzheimer. It is an irreversible, progressive syndrome characterized by neurocognitive impairment. Alzheimer’s Disease can affect both the elderly and the young, and its prevalence increases with age [1,2]. Approximately ten percent of the U.S. population age 65 and above has been diagnosed with AD [3]. It is the most common cause of dementia and contributes to 60–80% of cases. In 2024, an estimated 6.9 million Americans were living with AD [3]. With the aging of the baby boomer generation, the number of people with dementia is expected to increase dramatically in the coming years and is projected to reach 13.8 million by the year 2050 [4].

Clinically, AD patients present with a cluster of neuropsychological signs and symptoms, including memory loss, executive function impairment, and emotional and behavioral disturbances. As the severity of dementia progresses, it continues to undermine patient’s ability to live independently and causes burden to their families, caregivers, and communities. Thus, AD poses a threat to our aging society, family support systems, and medical and social services.

For decades, neuroscience research has heavily focused on seeking to understand the primary mechanism of AD and searching for pharmacological approaches for the treatment of dementia. We reviewed the current U.S. Food and Drug Administration (FDA) approved pharmacological treatment protocol for AD. We utilized PubMed, Google Scholar, the Alzheimer’s Association, and FDA news and events, and searched the literature for works including human drugs within primary sources of the literature references. We traced these sources back to the original trials and any related publications to find sources that mention FDA-approved monoclonal antibodies. We also traced these sources back to published trials and original publications to find sources that mention additional monoclonal antibodies. We utilized ClinicalTrials.gov and used the keywords monoclonal antibodies, dementia, and Alzheimer’s Disease to track past and currently active trials as well. For the topic of symptomatic pharmacological management, we also utilized PubMed to search for different trials that investigated its utility in practice.

## 2. Pathophysiology

### 2.1. Amyloid Cascade Hypothesis

The neuropathology of Alzheimer’s Disease is characterized by two specific features: extracellular amyloid plaques and intracellular neurofibrillary tangles (Figure 1). The National Institute on Aging and Alzheimer’s Association unified the diagnostic recommendations for AD by recognizing beta-amyloid and neurofibrillary tau deposits as unique entities that define this distinct neurodegenerative disease [5]. Amyloid precursor proteins (APPs) are polypeptide transmembrane proteins that are naturally present in most tissues [6]. They undergo cleavage by different secretases. Alpha-secretase cleaves APPs into a soluble peptide. However, through beta-secretase and subsequently gamma-secretase, the amyloid beta monomers Aβ1-40 and Aβ1-42 are generated, which then form oligomers and large polymers. Amyloid beta has a propensity to form protofibrils and fibrils and arrange these into insoluble beta-pleated sheets [7]. Amyloid beta exists in several forms, such as monomers, oligomers, protofibrils, and fibrils. Oligomers and protofibrils are neurotoxic [8,9]. The amyloid cascade hypothesis (ACH) stipulates that increasing amounts of aggregated extracellular Aβ, notably soluble oligomers, are neurotoxic, promote tau protein disintegration, and form neurofibrillary tangles; this leads to neuronal death, which is manifested as preclinical disease before eventually developing into dementia [10]. This ACH has sparked the development of monoclonal antibodies that work against amyloid beta proteins as disease-modulating agents for the treatment of AD.

The tau protein is a microtubule-binding protein which is responsible for axonal cytoskeleton stability and engages in microtubule assembly [12]. The hyperphosphorylation of tau proteins reduces their ability to bind to microtubules and forms insoluble filaments, also known as neurofibrillary tangles. Pathological NFTs interfere with synaptic signal transmission, disrupt mitochondrial function, promote oxidative injury, and alter genetic stability [13]. They occur in the hippocampal formation first and are later seen throughout the cerebral cortex [14]. The topographic spread of NFTs serves as a pathological marker for assessing AD severity. Amyloid beta accumulation precedes and likely triggers the formation of tauopathies [9]. Anti-amyloid beta monoclonal antibodies are developed to target different forms of amyloid and bind to fibrillar aggregates; these complexes are marked by monoclonal antibodies and will then be recognized by microglia, which leads to the phagocytic process [15]. Additionally, monoclonal antibodies also cause the dissolution of protein aggregates and create a disequilibrium between cerebral and peripheral circulation, leading to an efflux of beta proteins and a reduction in amyloid beta levels [16].

### 2.2. Synergistic Effect of Amyloid Beta and Tau Protein

In animal studies, Aβ plaques developed earlier than tau NFTs, blocking Aβ production stalled tau pathology formation [17,18]. Interestingly, the anti-Aβ antibody reduced early disease development but did not reduce late disease development with the presentation of tau pathology [19]. In addition to its induction within tau pathology, Aβ plaques also amplify tau phenotypes during the course of the disease [20]. This was corroborated by clinical observations where Aβ-enhancing genetic carriers had an early disease onset, yet the disease progression was comparable to sporadic disease [21]. This may imply that combining the therapy of anti-Aβ and anti-tau antibodies may be an alternative option for MCI or early AD treatment [20].

### 2.3. Symptomatic Treatment Basis

Regional losses of synapses and neurons accompanied by neurotransmitter deficits are observed in AD. The early and prominent neurodegeneration of the nucleus basalis of Meynert leads to decreased cholinergic neuronal projection. This synaptic failure results in the manifestation of memory and learning impairment [22]. This is the basis for symptomatic treatment with acetylcholinesterase inhibitors.

## 3. Disease-Modifying Therapy

Amyloid antibody medications have been surrounded by controversies since the initial approval of the first in the class—aducanumab. Novel monoclonal antibodies (MABs) slow disease progression and reduce cognitive decline compared to donepezil, which improves cognition [23]. However, they come with substantial side effects, such as amyloid-related imaging abnormalities (ARIAs). When patients and caregivers are confronted by this devastating disease, however, the maintenance of independence appears to be the most relevant and highly valued intervention as the disease progresses. Hartz et al. conducted a longitudinal study that examined the effects of treatments with lecanemab or donanemab. A total of 282 subjects with either positron emission tomography (PET) or cerebrospinal fluid (CSF) who were assessed for amyloid beta and tau proteins were enrolled. The instrumental activities of daily living (IADLs) and basic activities of daily living (BADLs) were measured with the clinical dementia rating, the sum of boxes (CDR-SB). An extension of patient-independent functional time was observed in both the lecanemab-treated group (10 months, 95% CI 4–18 months) and the donanemab group (13 months, 95% CI 6–24 months) in patients of the low/medium tau group [24]. With the careful selection of patients and the close monitoring of side effects, people living with mild cognitive impairment (MCI) and mild AD dementia will likely experience meaningful clinical benefits that outweigh the risks these drugs.

### 3.1. Aducanumab

Over the last decade, aducanumab has been one of the most promising drugs for Alzheimer’s Disease. It received controversial accelerated FDA approval in June 2021, and it was the first in this class to be approved, despite contradicting data from two phase 3 clinical trials. Additionally, the FDA requested a phase 4 confirmatory study from its manufacturer, ENVISION. In April 2022, however, the Centers for Medicare and Medicaid Services (CMS) announced restrictions on aducanumab coverage to randomized controlled trials, further limiting access to aducanumab in clinical practice. In January 2024, its manufacturer—Biogen (Cambridge, MA, USA)—announced the discontinuation of aducanumab, citing reprioritization of the company’s resources while denying any safety or efficacy concerns.

Aducanumab is an amyloid beta-directed, high-affinity human IgG1 monoclonal antibody. It has selectivity for aggregated soluble beta-amyloid oligomers and insoluble beta-amyloid fibrils compared to non-pathogenic, possibly neuroprotective monomers, therefore reducing the burden of beta-amyloid plaques in the brain [25,26,27,28]. A phase 1b trial PRIME (study 103, NCT01677572) demonstrated a dose-dependent reduction in brain beta-amyloid plaques, and a slowing of the progression of cognitive decline in subjects with MCI and mild AD patients with a PET-confirmed presence of brain beta plaques [26]. Based on these encouraging positive findings, two identically designed phase 3 clinical trials, namely EMERGE (NCT02484547) and ENGAGE (NCT02477800), were launched and started enrollment in 2015 [29].

EMERGE and ENGAGE were identically designed, randomized, double-blind, placebo-controlled trials. Subjects were randomized at 1:1:1 ratio to low-dose, high-dose aducanumab, or placebo groups stratified by apolipoprotein E (*ApoE*) ε4 carrier status. The low-dose group consists of a target dose of 3 mg/kg (*ApoE ε4*+) or 6 mg/kg (*ApoE ε4*−), while the high-dose group consists of a target dose of 6 mg/kg (*ApoE ε4*+) or 10 mg/kg (*ApoE ε4*−). Although both trials showed a reduction in brain amyloid beta, an interim analysis conducted in late 2018 revealed that they were unlikely to meet the primary endpoint of efficaciously slowing cognitive decline. Shortly after, Biogen announced the early termination of both trials in March 2019. Further reanalysis, however, confirmed that the high-dose arm met the primary endpoint measured with CDR-SB as well as the secondary endpoints measured by MMSE (Mini-Mental State Examination), ADAS-Cog 13 (Alzheimer’s Disease Assessment Scale–Cognitive Subscale 13), and ADCS-ADL-MCI (Alzheimer’s Disease Cooperative Study—Activities of Daily Living Scale for use in Mild Cognitive Impairment) in the EMERGE study [30]. With this, the FDA granted accelerated approval to aducanumab in 2021 against the advice of the independent Peripheral and Central Nervous System (PCNS) Drugs Advisory Committee. However, the FDA requires a post-approval confirmatory trial of the clinical benefits [30].

Aducanumab is suitable for patients with MCI due to AD and mild AD with PET-confirmed brain amyloid. An expert panel was assembled leading to the publication of Appropriate Use Recommendations for aducanumab in 2021. The expert panel recommended the strict selection of patient populations to be those studied for efficacy and safety. Titration, at the highest dose, is recommended for maximizing the chance of reaching efficacy while closely monitoring ARIAs, which have a higher incidence in patients taking high doses of aducanumab. Surveillance magnetic resonance imaging (MRI) should be obtained prior to treatment imitation, during dose titration, and when new symptoms develop concerning ARIAs. Should symptomatic ARIAs or moderate–severe ARIAs develop, treatment should be interrupted or discontinued [31].

Over the last two decades, several trials investigating anti-amyloid beta antibodies failed to prove clinical efficacy in the treatment of AD [32,33,34,35]. When the study results and data were presented to the PCNS Drug Advisory Committee, the committee voted against aducanumab’s approval (10 against, one abstention), yet the FDA granted accelerated approval [36,37]. This was followed by the resignations of three FDA advisory panel members [38]. Later on, aducanumab was rejected by both the European agency and the Japanese Medicines agency [39,40]. The restriction implemented by the CMS was deemed as a corrective action of the controversial FDA approval by some. Nonetheless, with discontinuation from the manufacturer, aducanumab will not be available to AD patients. Although surrounded by controversies, aducanumab rekindled the hope for this class of drugs.

### 3.2. Lecanemab

Lecanemab is a monoclonal antibody against amyloid-beta peptide protofibrils, which carry more neurotoxicity than monomers or insoluble fibrils [41]. Lecanemab is a humanized monoclonal antibody, and it has a higher binding capacity to amyloid-beta peptide (Aβ) protofibrils compared to aducanumab [27]. It has been shown to alter pathological Aβ protofibril accumulation in astrocytes and reduce their toxicity [42]. The CLARITY AD (CT03887455) study was a phase 3 double-blinded, placebo-controlled, randomized clinical trial conducted in multiple centers, investigating the safety and efficacy of lecanemab in patients with early Alzheimer’s Disease. Subjects with MCI due to AD were enrolled with PET or CSF-confirmed amyloid positivity. The primary endpoint was measured with CDR-SB; the secondary endpoint was measured with ADAS-Cog 14, ADCOMS (AD Composite Score), ADCS-MCI-ADL, and amyloid burden on PET. The adjusted mean change of CDR-SB from baseline at 18 months was less in the lecanemab group compared to the placebo group (1.21 in the lecanemab group and 1.66 in the placebo group, difference −0.45; 95% confidence interval, −0.67 to −0.23; *p* < 0.001) [43]. Additionally, the amyloid level was below the threshold for amyloid positivity. Markers of amyloid, tau, neurodegeneration, and the plasma glial fibrillary acidic protein (GFAP) were reduced in the lecanemab group, with the exception of NFTs [43].

With these confirmatory clinical benefits, the PCNS Drugs Advisory Committee convened and voted in affirmation of the clinical benefit of lecanemab for its indicated use. In July 2023, lecanemab became the first amyloid beta monoclonal antibody to receive traditional approval, following initial accelerated approval only in January 2023 [44,45].

Lecanemab dosing is based on actual body weight and is infused every two weeks. The confirmation of amyloid beta presence pathology prior to treatment initiation is required. Due to the higher risk of ARIAs, *ApoE* ε4 status prior to treatment is recommended [43]. A total of 95% of the subjects who completed the CLARITY AD core study (18 months) chose to continue in the open-label extension study. Over the treatment course across three years, treatment with lecanemab continued to demonstrate a reduction in cognitive decline on the CDR-SB [46]. The safety results from both core and open-label extended studies are also reassuring with the most common adverse events being infusion-related reactions and ARIA-H, ARIA-E. ARIAs were mostly mild to moderate based on radiographic features. ARIA-E also occurred generally in the first six months of treatment [46,47].

### 3.3. Donanemab

Donanemab is an immunoglobulin (Ig) G1 monoclonal antibody that binds to the N-terminal truncated form of beta-amyloid, facilitating amyloid plaque removal through phagocytosis mediated by microglia [48]. In a phase 3, double-blinded, placebo-controlled, randomized, multicenter clinical trial (TRAILBLAZER-ALZ 2, NCT04437511), a total of 1736 subjects who had mild cognitive impairment/mild dementia with PET-confirmed amyloid and low, medium, or high tau pathology were enrolled. The subjects were randomized at a 1:1 ratio to receive donanemab and a placebo; additionally, participants who received donanemab were switched to the placebo in a blinded manner after the dosing completion criteria were met. The primary endpoint was measured by the integrated Alzheimer Disease Rating Scale (iADRS), and the secondary outcome measurements included the CDR-SB score. The donanemab-treated group showed slowed clinical progression at 76 weeks in the low/medium tau population (iADRS least-squares mean: donanemab treated group −6.02, placebo group −9.27; difference: 3.25 [95% CI, 1.88–4.62]; *p* < 0.001) and in the combined population (iADRS least-squares mean: donanemab treated group −10.2, placebo group −13.1; difference: 2.92 [95% CI, 1.51–4.33]; *p* < 0.001) [49].

With these findings, on 3 July 2024, donanemab became the third amyloid beta monoclonal antibody approved by the FDA for AD treatment. However, the FDA requires the manufacturer of donanemab to conduct a post-market registry study to track events, including deaths and ARIAs. The final safety report is expected by February 2037. Biannual reports are also required to be submitted to the FDA [50]. A phase 3 trial assessing the safety and efficacy of donanemab in prodromal AD and mild dementia with evidence of tau pathology is currently underway [51].

### 3.4. Other Amyloid Monoclonal Antibodies

Crenezumab is a humanized anti-Aβ monoclonal IgG4 antibody with a higher affinity for oligomeric amyloid beta and the promoted disaggregation of Aβ oligomers [52]. It also has minimal binding compacity to the vascular amyloid, which potentially reduces the risk of ARIA development [53,54]. Two parallel phase three randomized placebo-controlled double-blinded trials, CREAD and CREAD 2, were conducted to evaluate the efficacy and safety of crenezumab in the treatment of MCI to mild AD. A total of 813 subjects were enrolled. Baseline characteristics, including age, sex, ethnicity, education, and cognitive scales were balanced between the placebo and crenezumab groups. The primary outcome was measured by CDR-SB. Unfortunately, after an interim analysis of CREAD suggesting that the study was unlikely to meet the primary endpoint, both trials were discontinued prematurely [55]. Following study termination, data analysis did not confirm the efficacy of crenezumab at primary or secondary endpoints. However, its safety profile was consistent with rare ARIAs and a similar incidence of ARIAs in between groups [55].

Gantenerumab is a humanized IgG1 antibody. It induces cellular phagocytosis of Aβ deposits by recruiting microglia [56]. Registered in 2010, a phase 3 trial examining the efficacy of gantenerumab dosed at 105 mg and 225 mg every 4 weeks was discontinued after interim futility analysis; however, dose-dependent effects were observed during exploratory data analyses [34]. Subsequently, another two phase 3 trials were launched to investigate the efficacy of a higher dose of gantenerumab. The primary endpoint was measured with CDR-SB, and the secondary endpoints included the amyloid level as examined with PET between the treatment and placebo group. Although gantenerumab lowered amyloid plaque burden, there was no evidence it slowed clinical progression [33].

### 3.5. Amyloid-Related Imaging Abnormalities

Amyloid-related imaging abnormalities are a known side effect associated with amyloid-targeting antibodies. ARIAs include ARIA vasogenic edema and/or sulcal effusions (ARIA-E) and ARIAs with hemosiderin deposition (ARIA-H). The incidence of ARIA-E, ARIA-H, and any ARIA were 12.6%, 17.3%, and 21.5, respectively, as associated with lecanemab, and 24.0%, 31.4%, and 36.8%, respectively, as associated with donanemab in both the *ApoE*-*e4* allele carrier and non-carrier [49]. ARIAs have been observed to be dose-dependent and occur earlier in the course of treatment. Therefore, a low-dose initiation with slow titration has been recommended. In addition to the dosage level, the *ApoE-ε4* carrier status appears to be the most significant risk factor for the development of ARIAs. Patients with homozygotes *ApoE-ε4* alleles have a higher hazard ratio and incidence of developing ARIA-H compared to heterozygotes and non-carriers [43,57]. *ApoE* genotyping can be considered. Patients with lobar microhemorrhages and superficial siderosis suggestive of underlying cerebral amyloid angiopathy (CAA) carrying *ApoE-ε4* have a particularly high risk of developing ARIAs [58,59].

In patients who developed ARIAs during the lecanemab clinical trials, most cases were asymptomatic, with mild to moderate radiographic severity. Symptomatic cases range from 6.1% to 39.3%, depending on the specific agents [43,49,60]. Severe or fatal symptoms could occur, as well as recurrent episodes of ARIAs. If clinical symptoms present, they vary from headache, disorientation, dizziness, gait disturbance, and focal neurologic deficits, to seizures. The resolution of ARIAs occurred within four months in the majority of cases treated with lecanemab [43].

The FDA has guidelines defining mild, moderate, and severe ARIA-E and ARIA-H [61]. Additionally, the Barkhof Grand Total Scale has been proposed to grade the severity of ARIAs [62]. For asymptomatic mild ARIA-E and mild ARIA-H, symptomatic mild ARIA-E, treatment can be continued based on clinical judgment [59]. For symptomatic moderate or severe ARIA-E, asymptomatic moderate ARIA-H, and severe ARIA-E, lecanemab should be withheld with MRI follow-ups for resolutions; the resumption of treatment should be based on clinical scrutiny and judgment. For severe ARIA-H, lecanemab should be withheld with MRI follow-ups for stabilization or resolution; with the resumption or discontinuation of treatment depending on clinical judgment. Given the limited understanding of ARIAs and the options for treatment, it is crucial to thoroughly counsel patients and caregivers regarding the risks individually.

### 3.6. Anti-Tau Protein Monoclonal Antibodies

Semorinemab is an anti-tau IgG4 antibody targeting the N-terminal domain of tau with high affinity [63]. It reduced tau protein toxicity in animal studies. In a phase 2 randomized, double-blinded, parallel design, placebo-controlled clinical trial, semorinemab did not show evidence of slowing clinical progression when measured with CDR-SBS; although it has an acceptable safety profile [64]. A second phase 2 trial only demonstrated its significant effect on cognition measured by ADAS-Cog11, mainly on memory, and again did not show evidence of slowing disease progression [65].

Tilavonemab is another IgG4 monoclonal antibody that works against tau. It also binds to the N-terminal domain of the human tau of the soluble extracellular tau in the brain. It is proposed that it may block extracellular tauopathies from spreading in the brain [66]. In a phase 2 randomized, placebo-controlled trial, tilavonemab did not show evidence of slowing disease progression in early AD patients. A higher dose was also included, and it appears there would not be dose-dependent clinical benefit either [67].

Gosuranemab is also an IgG4 monoclonal antibody that binds to N-terminal tau monomers and fibrils with high affinity [68]. Preclinical trials demonstrated the robust removal of N-terminal tau from cerebrospinal fluid and further reduced tau aggregation in cells. Gosuranemab was first evaluated in patients with progressive supranuclear palsy in a phase 2 trial (PASS-PORT trial) without evidence of slowing disease progression [69]. It was later studied in patients with early AD, which, again, did not reveal clinical evidence of delaying disease progression [70].

A meta-analysis pooling six randomized clinical trials assessing anti-tau protein monoclonal antibodies for AD concluded that semorinemab and tilavonemab demonstrated relatively good efficacy based on MMSE and ADAS-Cog, as well as the safety profile. However, there was no evidence of these drugs hindering disease progression [71].

## 4. Cognitive Medications

Cholinesterase inhibitors and NMDA receptor antagonists are FDA-approved medications for managing cognitive symptoms. The majority of patients with newly diagnosed MCI or mild AD are offered a trial of these medications. As there is no concrete evidence to suggest an alteration of the disease trajectory or neuroprotectivity, the duration of these medical therapies largely depends on patients’ clinical response and side effects.

### 4.1. Cholinesterase Inhibitors

Acetylcholinesterase (AChE) inhibitors include donepezil, rivastigmine, and galantamine. By inhibiting cholinesterase at the synaptic cleft, they increase cholinergic transmission and lead to symptomatic benefits. The expected benefit is modest, however, and there is a lack of evidence that they also exert neuroprotective effects or interfere with disease progression. Additionally, for patients receiving AChE inhibitors for 6 to 12 months, there were no significant changes in AD biomarkers such as CSF amyloid beta 1-42, or phosphorylated tau [72]. Therefore, they are considered as symptomatic therapies.

Three cholinesterase inhibitors are available, including donepezil, galantamine, and rivastigmine, all of which have demonstrated efficacy, with no major differences in efficacy or tolerability between these drugs, and the discontinuation rates were also similar among these three drugs. Donepezil is a once-daily medication that is also available in disintegrating tablets and transdermal patches. Galantamine is a twice-daily tablet or once-daily extended-release capsule. Rivastigmine is available as a tablet and transdermal patch. The most common side effects of cholinesterase inhibitors are gastrointestinal symptoms, including anorexia, nausea, and diarrhea. This often resolves with time or dose adjustment as it is usually dose-dependent. Anorexia can further lead to weight loss. Symptomatic bradycardia could occur, owing to their vagal tone-enhancing effect, especially when taken along with medications altering atrioventricular nodal conduction, or in patients with underlying conduction abnormalities [73,74]. Donepezil does not require renal or hepatic adjustment; however, galantamine should be avoided in patients with end-stage renal disease or severe hepatic injury. Additionally, galantamine is associated with increased mortality in MCI patients compared to a placebo, although it is still lower compared to expected rates in this population or in patients with AD; the causes of death included sudden death, heart attack, and suicide [75].

The duration of cholinesterase inhibitors is dependent mainly on the patient’s response and side effects. As the response can be slow and subtle, adequate trial time should be allowed before a decision on discontinuation is made. Cognition tests should be conducted at follow-up visits to track progression. In a randomized trial comparing donepezil, memantine, combined memantine, and donepezil for moderate to severe AD, the patients who continued on donepezil or memantine had higher SMMSE and lower BADLS, indicating less impairment compared to groups that were discontinued after three months of treatment; there was no difference of efficacy between donepezil and memantine [76]. If discontinuation is implemented, a tapering process should be implemented with close monitoring over 1 to 3 months for significant deterioration; should this occur, therapy should be reinstated [77]. Although, full recovery may not be achieved after discontinuation for more than six weeks [78].

### 4.2. NMDA Receptor Antagonist

Glutamate is an excitatory neurotransmitter that binds to inotropic noncompetitive N-Methyl-D-aspartate (NMDS) receptors and facilitates learning, memory, and neuroplasticity. The overexcitation of these neurons is observed in AD. NMDA receptor antagonists regulate calcium and signal transmission at synapses and improve cognition [79]. Memantine is an uncompetitive NMDA receptor antagonist. By binding to NMDA receptors, it reduces calcium influx and prevents neuronal overstimulation [80]. It reduces clinical deterioration in moderate-to-severe Alzheimer’s Disease measured by the CIBIC-Plus (Clinician’s Interview-Based Impression of Change Plus Caregiver Input) and the ADCS-ADLsev (Alzheimer’s Disease Cooperative Study Activities of Daily Living Inventory modified for severe dementia) [81]. A combination of donepezil and memantine to treat moderate to severe AD patients led to better cognitive functions in a meta-analysis [82]. It is generally well tolerated when taken orally and does not have significant side effects. Common side effects are headache, confusion, dizziness, constipation, urinary infections, and agitation [83].

## 5. Pharmacological Treatment for Neuropsychiatric Symptoms

Neuropsychiatric symptoms, including agitation, delusions, hallucinations, depression, and sleep disturbance are commonly seen in patients with AD [84]. They can present in the early stage of cognitive decline, and progress along with disease severity [85]. Neuropsychiatric symptoms can arise from underlying causes, including infection, medication toxicities, or metabolic derangements; therefore, a careful evaluation should be conducted before attributing the symptoms to dementia progression and initiating any medical management.

### 5.1. Depression

Roughly one-third of mild to moderate dementia patients suffer from major depressive disorder [86]. Selective serotonin reuptake inhibitors (SSRIs) such as citalopram have been recommended for the treatment of depression in patients with AD. Citalopram has been shown to reduce agitation and caregiver distress, although QT prolongation and potentially adverse effects on cognitive function limit its use in practice [87]. Sertraline is an alternative should citalopram is not tolerated. Tricyclic antidepressants are generally avoided due to their anticholinergic properties which worsen cognitive function. Clinical trials examining SSRIs have yielded mixed results. Some early trials have demonstrated the efficacy of citalopram but not sertraline for dementia patients with depression [88,89,90,91]. A recent systematic review and meta-analysis did not find a significant clinical effect of antidepressants as a treatment for dementia patients with depression, which adds controversy to the medical management of depression in this patient population [92].

### 5.2. Agitation and Aggression

Dementia-associated agitation and aggression are also common yet challenging to manage symptoms. While non-pharmacological management is preferred in this patient population, antipsychotics may be considered in patients afflicted with severe symptoms or as a refractory option to alternative treatment. Antipsychotics are associated with an increased risk of stroke, myocardial infarction, and mortality in patients with dementia. Therefore, the treatment of psychosis in dementia patients with antipsychotics is not recommended routinely [93]. A balance between potential benefits and harms should be reached when considering antipsychotics. The initiation of antipsychotics should align with the goal of care and the preference of the patients or their medical decision makers [93]. Risperidone is approved for use for up to six weeks for severe aggression in Europe. Atypical antipsychotics have been the agents of choice; however, these have been used in off-label circumstances. Additionally, the prolonged use of antipsychotics is discouraged, as clinical trials only display very modest benefits and even no benefits when compared to placebos over six months [94]. Brexpiprazole, a partial serotonin 5-hydroxytryptamine (HT)1A agonist and dopamine D2 agonist, has become the first drug to be approved by the FDA for the treatment of agitation associated with dementia due to Alzheimer’s Disease [95,96]. Brexpiprazole also carries a black box warning that elderly patients with dementia-associated psychosis are at increased risk of death when treated with antipsychotics [95].

### 5.3. Other Medications Used for Neuropsychiatric Symptoms

There are other medications used for neuropsychiatric or behavioral symptom management. Mirtazapine has been studied as a treatment option for depression. However, clinical trials did not show effectiveness when compared to a placebo and were associated with an increased risk of adverse events [97]. Trazodone is well tolerated but has also been found to have no benefit in the treatment of agitation in AD patients when compared to a placebo and behavior management techniques; however, it is often used to aid sleep onset [98]. Some clinical trials and case reports have shown positive results of antiseizure medications, such as carbamazepine, valproic acid, gabapentin, and lamotrigine, used in AD-associated neuropsychiatric symptoms given their known mood-stabilizing features. However, subsequent trials or studies have produced contradicting results. Their roles in dementia-associated neuropsychiatric symptoms are uncertain. Melatonin and melatonin receptor agonists have proved to be effective for patients without AD; however, investigations into their efficacy in AD patients for sleep disturbances are still under investigation [99]. Orexins produced by the hypothalamus play a critical role in modulating the sleep–wake rhythm. Dual orexin receptor antagonists, including suvorexant, lemborexant, and daridorexant, have shown efficacy in the treatment of sleep disturbance [100]. Suvorexant has become the first drug in this class to receive FDA approval for treating sleep disorders in mild to moderate Alzheimer’s Disease [101].

## 6. Active Registered Clinical Trials

There are a few trials that are actively recruiting or that are planning to recruit patients with AD to evaluate the efficacy and safety of different agents (Table 1).

## 7. Conclusions

Three amyloid beta monoclonal antibodies have been approved by the FDA to treat MCI and mild AD. With the manufacturer’s discontinuation of aducanumab, only lecanemab and donanemab remain available to patients. The careful selection of patients and close clinical monitoring, including the surveillance with MRIs, are warranted to maximize the benefits of these novel treatments. The complex pathological pathways seen in AD likely warrant a combination therapy between amyloid beta monotherapy and tau pathology-targeting agents instead of monotherapy.

## Figures and Tables

**Figure 1 biomedicines-12-02636-f001:**
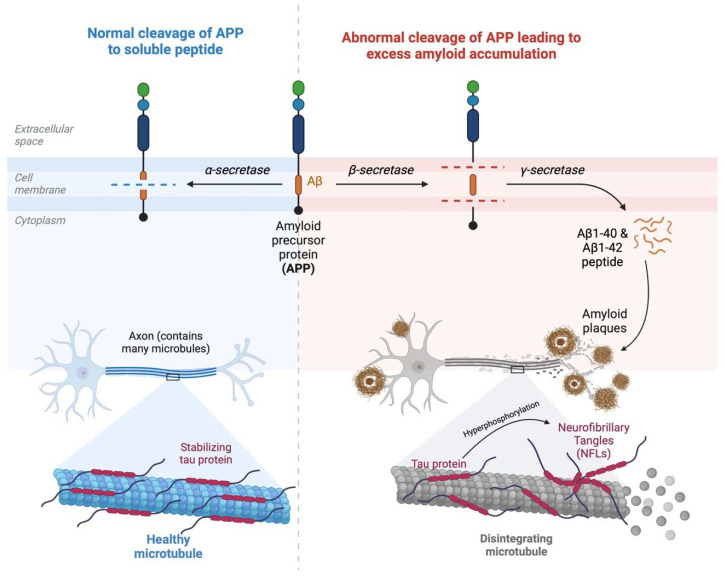
The pathophysiology of Alzheimer’s Disease [11]. On the left, in the normal cleavage of amyloid precursor proteins (APPs), α-secretase cleaves APPs into soluble peptides. The tau protein is a microtubule-binding protein responsible for cytoskeleton stability. On the right, in the abnormal cleavage of APPs, β-secretase and γ-secretase cleave APPs to produce Aβ1-40 and Aβ1-42, which further form protofibrils and fibrils and are arranged into insoluble amyloid plaques. Extracellular amyloid plaques are neurotoxic, leading to tau protein disintegration and, subsequently, neurofibrillary tangle formation.

**Table 1 biomedicines-12-02636-t001:** Active registered clinical trials.

Investigational Product/Clinical Trial ID	Proposed Mechanism of Action	Brief Summary	Study Design	Primary Measures
Tricaprilin [102] NCT05809908	An oral formulation of caprylic triglyceride. Induces a mild chronic ketosis to improve mitochondrial metabolism and boost cellular metabolism in AD by providing a fuel alternative to glucose	Evaluate the effects of tricaprilin on cognition, activities of daily living, resource utilization, safety, and tolerability in subjects with mild to moderate probable AD	Phase 3, randomized, double-blinded, placebo-controlled, parallel groups, multi-center	ADAS-Cog (26 weeks); ADCS-CGIC (26 weeks)
KarXT [103] NCT06585787	Xanomeline-trospium. Xanomeline is a muscarinic acetylcholine receptor activator. Trospium is a muscarinic receptor inhibitor that serves to block xanomeline’s peripheral actions	Evaluate the safety and efficacy of KarXT for the treatment of psychosis associated with Alzheimer’s Disease	A phase 3, randomized, double-blind, placebo-controlled, parallel group study	NPI-C: H + D (up to week 14)
Mastinib [104] NCT05564169	Oral tyrosine kinase inhibitor. Neuroprotective by inhibiting mast cell and microglia/macrophage activity	Evaluate masitinib as an adjunct to cholinesterase inhibitor and/or memantine in patients with mild-to-moderate Alzheimer’s Disease.	A multicenter, randomized, double-blind, placebo-controlled, parallel-group phase 3 Study	ADAS-Cog-11 (24 weeks); ADCS-ADL score (24 weeks)
AR1001 [105] NCT05531526	Selective inhibitor of phosphodiesterase 5. Increases the intracellular messenger cGMP, and possibly improves blood supply to the brain	Efficacy and safety of AR1001 over 52 weeks in participants with early Alzheimer’s Disease	A phase 3 double-blind, randomized, placebo-controlled, multi-center trial	CDR-SB (52 weeks)
Nilotinib BE [106] NCT05143528	Tyrosine kinase inhibitor (TKI) called Nilotinib BE (bioequivalent)	Evaluate the efficacy and safety of Nilotinib BE in subjects with early Alzheimer’s Disease	A multicenter, phase 3, randomized, double-blind, placebo-controlled study	CDR-SB (72 weeks)
AXS-05 [107] NCT05557409	Mimicks the extra-telomeric functions of the human telomerase reverse transcriptase (hTERT), inhibits neurotoxicity, apoptosis, and the production of reactive oxygen species induced by amyloid beta (Aβ) in neural stem cells	Evaluate the efficacy and safety of AXS-05 compared to a placebo for the treatment of agitation associated with Alzheimer’s Disease	Multi-center, double-blind, placebo-controlled, randomized study	CMAI [Up to 5 weeks]
GV1001 [108] NCT05303701	16-amino-acid peptide comprising a sequence from the human enzyme telomerase reverse transcriptase (TERT). Blocks Aβ oligomer-induced toxicity, and increases the survival of cells exposed to oxidative stress	Evaluate the efficacy and safety of subcutaneous administration of GV1001 1.12 Mg/day in patients with moderate to severe Alzheimer Disease	Multi-center, randomized, double-blinded, placebo-controlled, parallel design, prospective phase 3 study	SIB (24 weeks); CIBIC-plus (24 weeks)
Remternetug [109] NCT06653153	Monoclonal antibody targets a pyroglutamated form of Aβ	Evaluate the difference in time for the development or worsening of memory, thinking, or functional problems due to AD	Phase 3, randomized, double-blinded, parallel design	Time to Clinically Meaningful Progression as Measured by CDR (up to 255 weeks)
Cannabidiol [110] NCT06514066	Multifactorial. Anti-inflammatory and anti-excitotoxic barriers	Investigate the therapeutic benefits of cannabidiol (CBD) for dementia patients in Malaysia	Phase 2,3 randomized, double-blinded, cross assignment	ADAS-COG; NPI; PSQI; QOL-AD

ADAS-Cog, Change From Baseline in Alzheimer’s Disease Assessment Scale—Cognitive Subscale. ADCS-CGIC, Alzheimer’s Disease Cooperative Study—Clinical Global Impression of Change. NPI-C: H+D, Neuropsychiatric Inventory-Clinician: Hallucinations and Delusions. ADCS-ADL, Alzheimer’s Disease Cooperative Study Activities of Daily Living Inventory Scale. CDR-SB, Clinical Dementia Rating Scale—Sum of Boxes. CMAI, Cohen–Mansfield Agitation Inventory. CIBIC-plus, Clinician Interview-Based Impression of Change-Plus. SIB, Severe Impairment Battery. CDR, Clinical Dementia Rate. NPI, the Neuropsychiatric Inventory NPI scoring system. PSQI, Pittsburgh Sleep Quality Index. QOL-AD, overall quality of life. cGMP, Cyclic guanosine monophosphate.

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
