# Peer review of "Disease Modifying Monoclonal Antibodies and Symptomatic Pharmacological Treatment for Alzheimer’s Disease"

_biomedicines, 2024, doi:10.3390/biomedicines12112636_

Round 1

Reviewer 1 Report

Comments and Suggestions for Authors

The authors reviewed the recent pharmacological treatment for Alzheimer's disease. The authors focused on anti-amyloid therapy and cognitive medications. It is a well drafted manuscript with much detail discussed. The word is very well organized and structured. However, there are some major flaws in this review that need to be fixed before moving forward to publication.

1. The authors only reviewed three anti-amyloid therapy drugs. There are more drugs to be discussed and reviewed such as crenezumab from Roche and Gantenerumab. These, and some more treatments, should be discussed.

2. Aducanumab was proved not effective and discontinued. The authors briefly talked about this. The authors have to spend more effort in discussing in depth about the controversy and reason for getting discontinued. For the same reason drugs like Crenezumab should also be reviewed.

3. The whole paragraph that discussed the discontinue of Aducanumab has no citation.

4. The authors mentioned Lecanemab can decrease tau load. In the final discussion of this manuscript the authors argued that a combined therapy of amyloid and tau should be applied in the future. It seems that Lecanemab achieved both targets already. What are the authors' opinion on this?

Also this whole part 'Additionally, the amyloid level was below the threshold for amyloid positivity. Markers of amyloid, tau, neurodegeneration, and plasma glial fibrillary acidic protein (GFAP) were reduced in the lecanemab group, with the exception of NFLs. 

With these confirmatory clinical benefits, the PCNS Drugs Advisory Committee con- vened and voted in affirmation of the clinical benefit of lecanemab for the indicated use. In July 2023, lecanemab became the first amyloid beta monoclonal antibody to receive traditional approval, following the initial accelerated approval only in January 2023. Lecanemab dosing is based on actual body weight and is infused every two weeks. Confirmation of amyloid beta presence pathology prior to treatment initiation is required' has no reference.

5. The authors claimed that the combination of amyloid and tau targeting agents will be the future. Discussion like this with no supporting material is hollow and cursory. This review needs to discuss some tau-targeting drug such as Gosuranemab. 

Author Response

  1. added additional medications
  2. expanded controversies
  3. added citations
  4. added section in response to this and citations
  5. added section for anti-tau monoclonal antibodies

Reviewer 2 Report

Comments and Suggestions for Authors

In the manuscript entitled "Pharmacological treatment for Alzheimer’s Disease: A Review", Qi et al describe three monoclonal antibodies against amyloid beta for the treatment of mild cognitive impairment and mild AD. Following are the comments and questions that are aimed at improving clarity, adding depth to the mechanistic explanation, and addressing current challenges in AD research.

Below, I provide a list of several questions that need to be addressed by the authors:

1. The introduction of the amyloid-beta monoclonal antibodies (aducanumab, lecanemab, and donanemab) is a critical aspect of the manuscript. Could you provide more details or references to explain the specific mechanisms by which each of these monoclonal antibodies (aducanumab, lecanemab and donanemab) operates?

2. The focus on amyloid plaques is well justified, but the text could benefit from more explanation about why amyloid-targeted therapies have been prioritized, especially in light of recent debates about the amyloid hypothesis in Alzheimer’s research?

3. The mention of combination therapies targeting both amyloid and tau is intriguing. However, further elaboration on potential synergies between these therapies and existing drugs (e.g., acetylcholinesterase inhibitors, NMDA antagonists) could strengthen the manuscript. Can you expand on the rationale and evidence for pursuing combination therapies? Are there any preliminary results or ongoing trials evaluating these combinations?

4. While the manuscript mentions FDA approval, there is limited discussion about the outcomes of clinical trials and the efficacy of these treatments. What are the key results from the trials that led to the FDA approval of these antibodies? Are there any concerns about their long-term effectiveness or safety?

5. The section on future trials could be expanded to give a more comprehensive view of the research landscape. What specific research gaps exist in targeting both amyloid and tau pathologies? Can you provide more details about ongoing or planned trials addressing these areas?

There are a few typos and English and grammar errors that should be rectified.

Comments on the Quality of English Language

There are a few typos and English and grammar errors that should be rectified.

Author Response

  1. mechanism is embedded in the main paragraph
  2. added in pathophysiology section
  3. expanded on combination of amyloid and tau as treatment
  4. added long term registry studies available
  5. on going trials added

Reviewer 3 Report

Comments and Suggestions for Authors

The current study, “Pharmacological Treatment for Alzheimer’s Disease: A Review” did not cover the study. limited pharmacological agents were evaluated. 

·       The language needs major corrections and needs to be checked for grammar errors.

·    ·       Introduction: It should be detailed with references and citations or with many case examples in the section stating, "Disease can affect both the elderly and the young, and the prevalence increases with age." 

·   ·       Pathophysiology: The pathophysiology section was initially divided into two, but the information presented later is different from the beginning. The general information and mechanism part found in this section are very confusing. thus making the text difficult to understand. Also, the references are very inadequate. Review articles should have plenty of references and citations. However, the logic of writing an article has been followed here. Please check your reference list by reviewing multiple publications and studies on the subject.

·       Disease-Modifying Therapy: Couldn't other drugs have been chosen instead of the drugs named "The novel monoclonal antibodies"? Why was a review written specifically for only three drugs?

·       Also, "New monoclonal antibodies" need to explain for mechanism. However, the importance of monoclonal antibodies is not fully explained.

·       Aducanumab: "Based on these encouraging positive findings, two identically designed phase 3 clinical trials, namely 102 EMERGE(NCT02484547) and ENGAGE (NCT02477800), were launched and started enrollment in 2015." According to which publication or text was this sentence written?

·       References should be given for several essays mentioned in line 132.

·       Lecanemab: References should be provided for the Lecanemab section. For example, the study, FDA approval, and other treatments.

·       The molecular mechanism of the Lacenemab drug should be mentioned in detail. It is sufficient to give the abbreviation ARIA on lines 184 and 185. The long form of the abbreviation is given in the previous paragraphs.

Author Response

citations added;

pathophysiology section edited;

more disease modifying therapies added including anti-tau monoclonal antibodies;

mechanism in each section of the medications;

references added

spells, writing corrected

Round 2

Reviewer 2 Report

Comments and Suggestions for Authors

The author has addressed all the comments provided by reviewers, and now the manuscript is suitable for publication in a journal and can be accepted.

Author Response

Thank you for the review and support!

Reviewer 3 Report

Comments and Suggestions for Authors

The revisions are acceptable. There is only one issue: the title covers all treatment types, but the current article mainly covers antibody-based treatments. So please revise the title.  

Author Response

Comment: The title covers all treatment types, but the current article mainly covers antibody-based treatments.

Response: We have updated the title to "Disease Modifying Monoclonal Antibodies and Symptomatic Pharmacological treatment in Alzheimer's Disease: A Review" to reflect the emphasis on monoclonal antibodies treatment. Addtional symptomatic treatment inclduing AChEI, NMDA, neuropsychiatric are also discussed in this review, and title added "symptomatic pharmacological treatment" to accurately reflect this section.